# Decrease in the P2 Amplitude of Object Working Memory after 8 h-Recovery Sleep Following 36 h-Total Sleep Deprivation: An ERP Study

**DOI:** 10.3390/brainsci13101470

**Published:** 2023-10-18

**Authors:** Yongcong Shao, Ziyi Peng, Lin Xu, Jie Lian, Xin An, Ming-Yang Cheng

**Affiliations:** School of Psychology, Beijing Sport University, Beijing 100084, China; budeshao@bsu.edu.cn (Y.S.); pzyi121@163.com (Z.P.); rzxulin1997@126.com (L.X.); 2021210372@bsu.edu.cn (J.L.); axinlll@163.com (X.A.)

**Keywords:** sleep deprivation, recovery sleep, objective working memory, event-related potentials, n-back

## Abstract

The impact of sleep deprivation on working memory can only be reversed by recovery sleep (RS). However, there are limited electrophysiological studies on the effect of RS on the improvement in working memory after sleep deprivation, and the changes in the early components of event-related potentials (ERPs) before and after RS are still unclear. Therefore, this study aims to explore the effects of RS on the earlier ERP components related to object working memory following 36 h of total sleep deprivation (TSD). Twenty healthy male participants performed an object working memory task after 36 h of TSD and after 8 h of RS. Electroencephalogram data were recorded accordingly while the task was performed. Repeated ANOVA showed that P2 amplitudes related to object working memory decreased significantly after 8 h of RS compared to after a 36 h period of TSD, but there was no significant difference from baseline (BS), which indicates a trend of recovery to the baseline state. An 8 h RS can partially improve impaired object working memory caused by TSD. However, a longer period of RS is needed for the complete recovery of cognitive function after a long period of TSD.

## 1. Introduction

Adequate sleep is fundamental to physical and mental health [1,2]. Although individuals need seven or eight hours of sleep per night, people are always passively or actively undergoing sleep deprivation due to various reasons, such as heavy schoolwork, work pressure, mobile phones, and social media use. The resulting long-term lack of sleep has been found to disrupt the normal functioning of the body’s physiological system, which can lead to the development of metabolic disorders and problems in one’s endocrine and immune functions [3,4]. Lack of sleep can also seriously affect important mental functions, such as cognition, which is detrimental to the consolidation and optimization of memory [5]. The COVID-19 outbreak in 2020 disrupted people’s daily routines and was a wake-up call to the whole world about the importance of health. According to a report released by the Chinese Sleep Research Society, the onset of the COVID-19 pandemic has brought about a delay in bedtime onset by two or three hours, and the incidence of insomnia among Chinese adults reached as high as 38.2%, suggesting that over 300 million Chinese individuals are suffering sleep disorders [6]. Therefore, studying the impact of sleep deprivation on cognitive function can help people better understand sleep and avoid the safety hazards caused by sleep deprivation. This research also has application value for the optimization of cognitive performance in occupations with high night-work requirements.

Working memory is a limited system of the brain that temporarily stores and processes information [7]. As an important subcomponent of executive function, working memory is the basis of many advanced cognitive processes [8,9]. Numerous studies have shown that sleep deprivation significantly impairs working memory. For instance, a number of studies determined that compared with baseline (BS), 36 h of total sleep deprivation (TSD) resulted in a significant decrease in working memory task performance as evidenced by prolonged response times and decreased accuracy [8,10,11]. Functional magnetic resonance imaging (fMRI) studies have also found that after TSD, the activation of the posterior parietal lobe decreases [12,13,14]. Since this region is involved in the retrieval operations of working memory, its reduced activation may be associated with decreased performance after TSD [15]. In contrast, the increased activation of the anterior cingulate gyrus after TSD may be associated with an increase in performance errors [8,16]. The study conducted by Lythe et al. has unveiled a bilateral frontoparietal load response network engaged during n-back tasks under conditions of both standard rest and TSD [17]. Notably, heightened cognitive demands during high-load working memory tasks corresponded to a reduction in activity within the right ventrolateral prefrontal cortex. Conversely, in the context of low-load working memory tasks, activity within the right parietal cortex exhibited augmentation. Notably, the degree of activation within these two regions during typical resting states offered predictive insights into their responses during periods of sleep deprivation. This finding suggests the potential for anticipatory characterization of neuronal effects resulting from sleep deprivation based on patterns of activation observed during normal resting states. Choo found that after TSD [8], complex cognitive tasks would lead to neural protection and compensation due to the overactivity of the prefrontal cortex and thalamus. Given the important role of the thalamus and related brain regions in cortical activation, changes in their functional connectivity can predict the impairment of working memory tasks after TSD [18]. The significance of working memory becomes notably apparent considering the findings by Lei et al. [19]. Following a period of 36 h TSD, their study illuminated a critical observation: An aberrant correlation emerged between the significant network and the default network. This unforeseen relationship hindered the optimal allocation of cognitive resources, thereby impeding the brain’s ability to effectively engage task-relevant regions precisely when their activation was required. This underscores the pivotal role that working memory plays in maintaining cognitive processes, particularly under conditions of sleep deprivation.

Event-related potentials (ERPs) provide an objective means of quantifying cognitive processes due to their non-invasive, reproducible, and quantifiable nature. Consequently, they offer distinctive merits within psychophysiological measurement and stand as reliable tools for delivering spatiotemporal insights into cognitive processing dynamics [20]. Mecklinger and Pfeiffer showed that the function of the phonological working memory is related to the changes in negative slow waves in the left frontal area and bilateral positive slow waves in the parietal area, as well as the fact that spatial working memory is related to negative slow wave changes in the posterior parietal area [21]. Compared to non-sleep deprivation, 36 h of TSD decreased the amplitude of N2-P3 components related to working memory and prolonged latency [10,11,22,23]. This may be because sleep deprivation impairs cognitive resources and sustained attention during working memory processing, thus leading to decreased decision-making ability in the cognitive matching response [10] Early ERP components, such as N1 and P2, can reflect the changes in the sensory coding process as well as attention processing in relation to working memory before and after TSD [23,24]. Although many studies have used ERP technology to explore the effects of sleep deprivation on cognitive function, only a few studies have measured changes in early ERP components before and after TSD, as well as the effect of recovery sleep (RS) on early ERP components.

Working memory exhibiting functional decline due to sleep deprivation requires RS to repair it [25,26]. However, there are only a limited number of electrophysiological studies on the effect of RS on working memory after TSD. Furthermore, changes in the early components of ERP before and after RS are still unclear. Therefore, this study aims to explore the effect of RS after a period of TSD on the earlier ERP components related to object working memory.

## 2. Materials and Methods

### 2.1. Participants

A cohort of forty healthy young males, ranging in age from 20 to 27 years (mean age = 23.5 years), was selected as participants for this study. Participants were randomly assigned to two groups: the nocturnal sleep (NS) group (20 participants) and the SD group (20 participants), (See Table 1). Recruitment was conducted via social media platforms, and a rigorous screening process was employed to ensure the eligibility of all candidates. All participants were right-handed and either had normal vision or corrected vision. All participants had a Pittsburgh Sleep Quality Index < 5 points [27]. The participants had no history of psychoneurotic disease, no recent symptoms of acute infection, and no recent history of medication. All participants were asked to refrain from smoking, drinking alcohol/coffee, and taking any drugs for at least 48 h prior to the experiment, and were instructed to maintain a normal sleep pattern for 1 week. Participants were informed of the process and potential risks of sleep deprivation prior to the start of the experiment. Participation in the study was voluntary. The participants were then provided with written informed consent. They were then paid after the experiment. The study was approved by the Ethics Committee of Beihang University and has been carried out in accordance with the Declaration of Helsinki.

### 2.2. Experimental Task

The experimental task was an object working memory test. The classical visual n-back task was used to probe the object working memory subsystem, and the 2-back object working memory task was selected (see Figure 1). Before the stimulus appeared, a 200 ms fixation point was first presented. Each stimulus was presented for 400 ms, and the stimulus onset asynchrony time was 1600 ms. There were 12 kinds of stimuli in the experiment—all of which were geometric figures—with a viewing angle of 2° × 2°. Each participant conducted a total of 122 trials. The experimental task required the participants to judge whether the stimulus currently presented was consistent with the second stimulus presented before it. If consistent, the current stimulus would be the target stimulus. If the stimulus was the target stimulus, the participants would click the left mouse button with the right index finger; if it was a non-target stimulus, they would click the right mouse button with the right middle finger. The target and non-target stimuli appeared in the same ratio at random. The test time was approximately 5 min.

### 2.3. Experimental Procedures

Before the experiment, participants were familiarized with the entire experimental procedure by performing the object working memory task until they had achieved a 90% accuracy rate. The SD group and NS group entered the laboratory at 18:00 on the day before the experiment for preparatory work. The NS group performed the first task at 20:00 (baseline state; NS-BS). Following this, the participants slept in the laboratory that night (ensuring a sleep time of at least 8 h), woke up the next day at 7:00, and performed the second task at 8:00 (0 h sleep deprivation; NS-SD0) with EEG data being recorded simultaneously.

The SD group slept all night in the laboratory and underwent sleep monitoring by sleep bracelets (ensuring a sleep time of at least 8 h). On the first day of the experiment, participants were woken up at 8:00, and TSD began. This continued for 36 h until the TSD ended at 20:00 the following night. At these two time points, the participants took the object working memory task and had their electroencephalogram (EEG) data recorded. The period of RS began at 23:00 on the second day and ended at 7:00 on the third day. The participants were then made to perform the object working memory task again at 8:00, with their EEG data being recorded concurrently (see Figure 2). During the TSD period, participants were only allowed to engage in non-strenuous activities, such as talking, reading, playing games, and using computers. Participants could move freely and perform light physical activities, but they could not leave the laboratory. Smoking and drinking stimulant drinks (e.g., coffee, tea, cola) were forbidden during the study. The experiment was performed on two participants at a time. The researchers accompanied and observed the participants throughout the entire duration of the experiment to ensure that they stayed awake for the whole TSD period.

### 2.4. EEG Data Collection and Recordings

The Neuroscan ERP system (Charlotte, NC, USA) was used to record the EEG signals by means of a 64-channel electrode cap and the international 10–20 system (see Figure 3). The SCAN system served as the core and was combined with a SynAmps2 amplifier. The reference and grounded electrodes were then placed on the forehead of the participants. These electrodes were placed on the lateral side of both eyes to record the horizontal EEG reading, while electrodes were placed above and below the left eye to record the vertical reading. With the bilateral mastoid as the reference electrode, the analog filter was 0.05–17 Hz. The sampling frequency was 1000 Hz, and the electrode impedances were maintained below 5 kΩ.

### 2.5. Data Analysis

Due to technical issues, data from one participant in the NS group was excluded during post-processing (NS group: 19; SD group: 20).

Behavioral data include mean correct response time, correct response rate, and number of correct per unit of time. For the NS group, a dependent t-test for paired samples was used to determine the two sleep states (NS-BS, NS-SD0h). For the SD group, repeated measures ANOVA was used to determine the sleep state (BS, 36 h-TSD, 8 h-RS) main effects. For non-compliance with the sphericity test, the results were corrected with the Greenhouse-Geisser test.

After a continuous EEG recording was completed, the data were analyzed by the SCAN 4.3 (Neuroscan Products) program and eye-moment artifacts were corrected with ocular artifact reduction. Stimulus-locked ERP was baseline-corrected in the range of −100 ms to 0 ms before the start of the stimulus. The average electrooculogram (EOG) and electromyogram (EMG) readings with amplitudes exceeding ±100 µV were removed as artifacts. After data averaging, we selected a total of 900 ms for analysis (100 ms before stimulation to 800 ms after presentation). Since working memory is most closely related to the prefrontal lobe, we choose to use the arithmetic average of the three electrode points F3, Fz, and F4 to calculate the mean amplitude and latency of N1 (50–150 ms) and P2 (100–250 ms) [28]. For the NS group, a dependent t-test for paired samples was used to determine the two sleep states (NS-BS, NS-SD0h). For the SD group, repeated measures ANOVA was then used for all ERP outcomes to identify the main effects of sleep states (BS, 36 h TSD, and 8 h RS). Greenhouse-Geisser corrections for non-sphericity and Bonferroni post hoc tests were also performed.

## 3. Results

### 3.1. Behavior Performance

In the NS group, the analysis yielded results indicating that sleep state exerted no significant influence on reaction time (*p* = 0.44), the correct rate (*p* = 0.99), and the frequency of correct responses per unit of time (*p* = 0.55) (See Table 2).

In the SD group, the analysis yielded results indicating that sleep state exerted no significant influence on reaction time (F (2,38) = 0.723, *p* = 0.492, η^2^ *p* = 0.037). However, the sleep state exhibited a substantial impact on the correct rate (F (2,38) = 11.579, *p* = 0.001, η^2^ *p* = 0.382) and the frequency of correct responses per unit of time (F (2,38) = 4.813, *p* = 0.014, η^2^ *p* = 0.202). Upon closer examination using post hoc analyses, notable patterns emerged. Specifically, following 36 h of sleep deprivation, a significant reduction was observed in both the percentage of correct responses (*p* = 0.002) and the frequency of correct responses per unit of time (*p* = 0.024). Conversely, after undergoing restorative sleep, a considerable increase was noted in both the percentage of correct responses (*p* = 0.001) and the frequency of correct responses per unit of time (*p* = 0.025) when compared to the baseline measurements (See Table 3).

### 3.2. Amplitude

In the NS group, the analysis of the results reveals that the sleep states had no significant impact on N1 and P2 (*p* = 0.85; *p* = 0.32), as presented in Table 3. In the SD group, the analysis of the results reveals that the sleep states had no significant impact on N1 (F (2,38) = 2.073, *p* = 0.140, η^2^ *p* = 0.098) as presented in Table 4 and Figure 4. In contrast, a significant main effect of sleep states on P2 was observed (F (1.461,27.763) = 5.938, *p* = 0.013, η^2^ *p* = 0.238), also depicted in Table 5 and Figure 4.

Further examination of these effects using posterior comparisons elucidated notable trends. Specifically, in comparison to the baseline condition, following 36 h TSD, there was a noticeable elevation in P2 amplitude that approached significance (*p* = 0.055). Following RS, a substantial reduction in P2 amplitude was observed (*p* = 0.008), yet no significant difference emerged when compared to the baseline (*p* = 0.078).

### 3.3. Latency

In the NS group, the analysis of the results reveals that the sleep states had no significant impact on N1 and P2 (*p* = 0.51; *p* = 0.63). In the SD group, upon conducting statistical analysis, it was determined that the N1 latency under different sleep states did not yield a significant effect (F (2,38) = 0.595, *p* = 0.557, η^2^ *p* = 0.030). In contrast, a notable alteration in P2 latency emerged, manifesting as a statistically significant change (F (2,38) = 6.505, *p* = 0.004, η^2^ *p* = 0.255).

Further exploration using posterior comparisons unveiled distinct trends. Following 36 h TSD, a considerable reduction in P2 latency was observed, rendering the change statistically significant (*p* = 0.002). However, the variance in latency before and after RS did not reach significance (*p* = 0.212). In comparison to the baseline condition, the latency of P2 after RS exhibited a notable reduction that was statistically significant (*p* = 0.032).

## 4. Discussion

This study investigated the restorative effect of 8 h RS on the impairment of object working memory caused by 36 h of TSD. By using the 2-back experimental paradigm of object working memory, we observed that the P2 amplitude decreased significantly after 8 h of RS and that the P2 latency was significantly shorter than BS. However, the amplitude and the latency of N1 did not change significantly. Previous research has shown that after 36 h of TSD, the P2 amplitude may increase due to the influence of compensatory effects [11], which was confirmed by the results of this study. However, our study found that the compensatory effect was not persistent and that the restorative effect of RS did not increase the ERP component amplitude as much as previously thought but did manifest itself closer to the baseline level. Thus, our findings affirm that an 8 h duration of RS has the capacity to partially mitigate the effects of TSD on object working memory.

The principal discovery in this study centers on the marked alteration observed in P2 amplitude following 8 h of RS in contrast to TS. Notably, during this period, P2 latency demonstrated a significant reduction in comparison to baseline. This finding aligns with the work of Zhang et al. [10], who similarly observed a substantial shortening of P2 latency after RS in comparison to TSD, providing evidence of RS’s influence on P2 dynamics. P2, a widely examined ERP component associated with positive stimuli, notably represents the most prominent forward movement within the prefrontal lobe, peaking around 200 ms. P2 serves as a reflection of the attention-processing mechanism for perceptual cues [29], particularly in the context of visual feature analysis [30]. Moreover, P2 exhibits heightened sensitivity to shifts in task-related attention and the demands of working memory [31]. After TSD, the resources for sensory input and the stimulus-matching process are reduced. According to the cognitive demand-specific hypothesis [32], under specific cognitive needs, the brain may adaptively employ additional resources that are not used under normal resting conditions to perform certain cognitive tasks, thus explaining the P2 amplitude increase after TSD. However, this compensatory effect does not persist. After 8 h of RS, the perceptual matching function and the initial stages of attention regulation were partially restored. Participants were able to better concentrate when performing object working memory tasks, like identifying the geometric features of the presented figure more quickly and accurately and matching the currently presented geometric figure with the previous one more accurately. At the same time, the extra cognitive resources employed by the brain under stress gradually returned to normal levels. While the P2 amplitude decreased post-8 h RS compared to 36 h TSD, it did not significantly differ from the baseline. This result indicates that the restorative effect of RS on TSD may not always manifest as an increase in the ERP component amplitude but as a tendency to return to baseline level, whether there is a damaging effect or compensatory effect after TSD.

In the context of this study, N1 exhibited no statistically significant variance before and after RS. However, a discernible trend towards recovery was observed, marked by shortened latency. N1, significantly influenced by cognitive engagement, is primarily associated with attention and consciousness [33]. This subtle alteration in N1 potentially underscores the significant role of RS in the early phases of attention and sensory processing. TSD diminishes participants’ capacity to sustain attention across wakefulness and slumber, whereas RS ameliorates the latency in the attentional process. Consequently, RS may enable individuals to promptly respond to sensory coding processes, thus facilitating a more adept reaction to the sensory environment.

Studies have shown that the improvement effect of RS can also be reflected in the changes in ERPs’ late components [34]. Our study provides evidence for this type of conclusion from the perspective of the early components of EEG. The working memory task selected in this study is sensitive to sleep deprivation, so we can see the differences in related EEG components under different states. Working memory is the ability to temporarily store and manipulate information during cognitive tasks. To this day, working memory is considered the core foundation of human advanced cognitive activities [35]. The early components selected in this study reflect the early stages of attention and perception processing in working memory processing [36]. RS first improves the initial stage of attention regulation reflected by N1 and P2, which is further manifested in the shortening of the P2 latency, reflecting the recovery of the individual’s attention needed to process stimuli in the visual cortex [37]. Combined with the results of our research, it can be seen from the changes in the P2 components before and after RS that the improvement primarily occurs in the process of attention and sensory input. With the recovery of the sensory process, the increase in the potential amplitude of the P3 component after the P2 component may reflect the recovery of the participants’ ability to discern the target stimulus since P3 latency reflects the time window for stimulus classification and evaluation [38,39,40].

This study significantly contributes to the existing literature by shedding light on the impact of RS on working memory subsequent to total sleep deprivation TSD. It further elucidates alterations in early ERP components like N1 and P2 under these conditions. However, several limitations warrant acknowledgment in our study. Firstly, the study’s participant pool was exclusively composed of young males. Consequently, caution is necessary when extrapolating findings to diverse populations, such as women, children, or the elderly. Secondly, the influence of circadian rhythms, which display individual variability [41,42,43], on behavior must be acknowledged. ERP data after TSD were collected at varying time points: (1) at 8:00 on the first day, (2) at 20:00 on the second day of the experiment, and (3) at 8:00 on the third day after the 8 h RS. Variations in collection times could potentially impact experimental outcomes. Thirdly, we did not use a separate anxiety questionnaire to assess the participants. Lastly, the study solely employed a 2-back working memory task without encompassing a broader range of working memory loads. Turner et al. argued that low-load working memory leans heavily on attention, while high-load working memory hinges more on executive control [44]. Drummond et al. demonstrated that task complexity influences the brain’s compensatory response to complete TSD [32]. Future investigations could manipulate diverse working memory loads to delve deeper into RS’s effects on executive function and other higher-order cognitive activities following TSD. Additionally, coupling traceability analysis and fMRI technology could enhance the precision of pinpointing changes within relevant brain regions, thus compensating for ERP’s limited spatial resolution. We did not investigate the differences in sleep architecture during normal and RS nights, necessitating further studies in this regard.

In summary, our study offers ERP-based evidence that 8 h of recovery sleep (RS) can partially ameliorate the detrimental effects on object working memory induced by 36 h of total sleep deprivation (TSD). This restorative influence predominantly surfaces in the enhancement of early sensory processing stages, as well as the bolstering of perception matching and attention regulation functions. Nonetheless, it’s worth noting that RS falls short of completely reinstating the cognitive decline stemming from the 36 h TSD. Further investigations are warranted to delineate the optimal duration of RS required to fully restore working memory to an optimal state.

## Figures and Tables

**Figure 1 brainsci-13-01470-f001:**
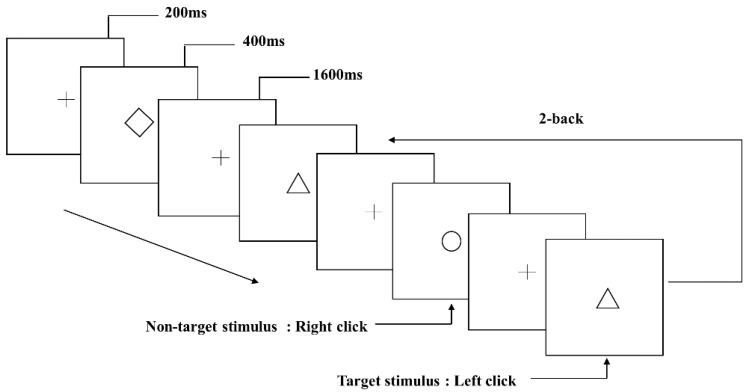
Schematic diagram of object working memory task.

**Figure 2 brainsci-13-01470-f002:**
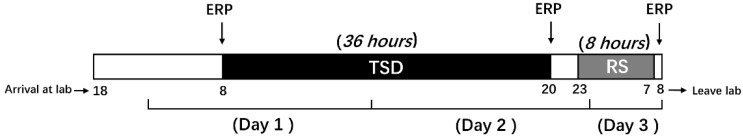
Experimental design. Participants had 8 h-RS after 36 h-TSD. TSD: total sleep deprivation.

**Figure 3 brainsci-13-01470-f003:**
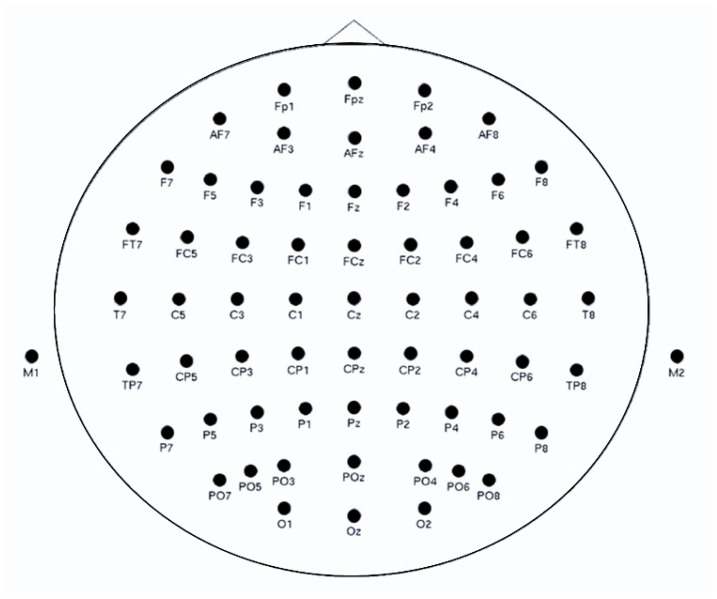
64-channel EEG electrode location map.

**Figure 4 brainsci-13-01470-f004:**
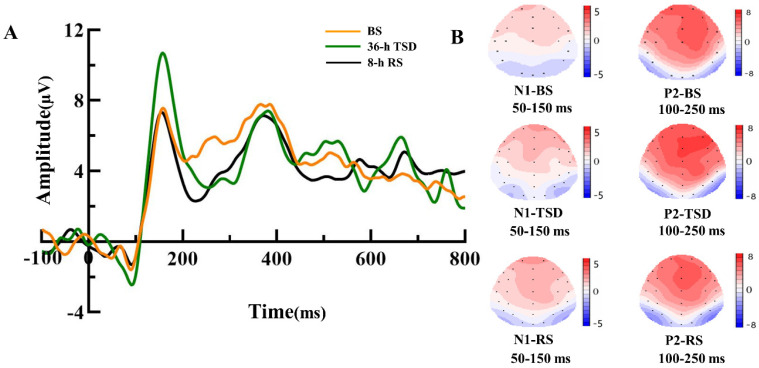
(**A**) BS (orange), 36 h-TSD (green), and 8 h-RS (black) ERP amplitude of the correct response of the object’s working memory. The effect of sleep state on N1 and P2. The data were averaged from F3, Fz, and F4. (**B**) Topographic map of the correct response in the working memory task in different sleep states. BS: baseline; TSD: Total sleep deprivation; RS: recovery sleep.

**Table 1 brainsci-13-01470-t001:** Demographic information of each group (mean ± standard deviation).

Group	Nocturnal Sleep	Sleep Deprivation
Number	20	20
Gender	Male	Male
Age (y)	23.55 ± 2.33	23.11 ± 2.40
PSQI	3.54 ± 0.93	3.66 ± 0.50

**Table 2 brainsci-13-01470-t002:** Performance data on the 2-back task in the nocturnal sleep (NS) group (n = 19, M ± SD).

	NS-BS	NS-SD0
Mean Reaction time	519.83 ± 117.88	538.57 ± 134.62
Correct rate	0.89 ± 0.07	0.89 ± 0.06
Correct number/s	1.83 ± 0.47	1.79 ± 0.49

BS: baseline. SD: sleep deprivation.

**Table 3 brainsci-13-01470-t003:** Performance data on the 2-back task in in the sleep deprivation (SD) group (n = 20, M ± SD).

	BS	36 h-TSD	8 h-RS
Mean Reaction time	522.19 ± 88.44	543.85 ± 107.88	526.07 ± 80.19
Correct rate	0.88 ± 0.09	0.79 ± 0.13 *	0.87 ± 0.07
Correct number/s	1.76 ± 0.44	1.53 ± 0.45 *	1.69 ± 0.34

BS: baseline. TSD: Total sleep deprivation. RS: recovery sleep. * *p* < 0.05 vs. baseline.

**Table 4 brainsci-13-01470-t004:** The mean amplitude and latency of the N1 and P2 components in the correct response condition across multiple electrode sites at baseline and after 0 h of sleep deprivation (SD0) in the nocturnal sleep (NS) group (n = 19, M ± SD).

	Amplitude		Latency	
	N1	P2	N1	P2
NS-BS	−2.05 ± 2.82	8.07 ± 2.15	98.13 ± 18.98	183.63 ± 20.95
NS-SD0	−1.95 ± 2.23	7.40 ± 2.90	94.66 ± 17.12	179.87 ± 26.47

BS: baseline. SD: sleep deprivation.

**Table 5 brainsci-13-01470-t005:** The mean amplitude and latency of the N1 and P2 components in the correct response condition across multiple electrode sites in three sleep states in the sleep deprivation (SD) group (n = 20, M ± SD).

	Amplitude		Latency	
	N1	P2	N1	P2
BS	−2.89 ± 1.62	8.63 ± 4.99	92.88 ± 20.43	182.28 ± 23.26
36 h-TSD	−4.40 ± 3.63	11.91 ± 6.50	93.41 ± 12.01	160.88 ± 15.72 *
8 h-RS	−3.25 ± 2.69	7.15 ± 4.90	97.12 ± 20.46	165.20 ± 14.29 *

BS: baseline. TSD: Total sleep deprivation. RS: recovery sleep. * *p* < 0.05 vs. baseline.

## Data Availability

The datasets generated for this study are available on request to corresponding authors.

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
