# Peer review of "Decrease in the P2 Amplitude of Object Working Memory after 8 h-Recovery Sleep Following 36 h-Total Sleep Deprivation: An ERP Study"

_brainsci, 2023, doi:10.3390/brainsci13101470_

Round 1

Reviewer 1 Report

The topic of cognitive impairment due to sleep deprivation is extremely relevant, primarily due to workers who have a night work schedule. In particular, this directly concerns medical workers who, after night duty, must continue their work day. The possible mistakes they make due to sleep deprivation can prove fatal to the patients. Therefore, it is important to understand the risks of sleep deprivation and how to combat them.

However, a number of comments arose during the review:

1) The authors need to more clearly define the inclusion/exclusion criteria from the study, as well as a detailed description of the study group (it makes sense to add a table).

2) Was testing for anxiety carried out before inclusion in the observation group? Because anxiety also affects sleep.

3) How was sleep deprivation achieved? Did the patients drink caffeinated drinks?

4) A large number of abbreviations are used in the figures and diagrams; for clarity, it is necessary to provide explanations for the abbreviations directly below the figures.

Author Response

Thank you for your letter and for the reviewers’ comments concerning our manuscript entitled “Decrease in the P2 Amplitude of Object Working Memory after 8h-Recovery Sleep following 36h-Total Sleep Deprivation: An ERP Study”. All of the comments were valuable and helpful for revising and improving our paper. We have studied comments carefully and have made revisions that we hope will meet with approval. The revised portions are marked in red in the paper. The main corrections in the paper and the responses to the reviewers’ comments are as follows:

Review#1

The topic of cognitive impairment due to sleep deprivation is extremely relevant, primarily due to workers who have a night work schedule. In particular, this directly concerns medical workers who, after night duty, must continue their work day. The possible mistakes they make due to sleep deprivation can prove fatal to the patients. Therefore, it is important to understand the risks of sleep deprivation and how to combat them. However, a number of comments arose during the review:

R1-1. The authors need to more clearly define the inclusion/exclusion criteria from the study, as well as a detailed description of the study group (it makes sense to add a table).

Response:

Thank you very much for your suggestion. We have supplemented the participant inclusion criteria, and presented demographic information in table 1(page 3).

“All participants were right-handed, and either had normal vision or corrected vision. According to the Pittsburgh Sleep Quality Index [27], all participants had good sleeping habits and a normal sleep-wake rhythm (Pittsburgh Sleep Quality Index <5 points). The participants had no history of psychoneurotic disease, no recent symptoms of acute infection, and no recent history of medication. All participants were asked to refrain from smoking, drinking alcohol/coffee, and taking any drugs for at least 48 h prior to the experiment, and were instructed to maintain a normal sleep pattern for 1 week.”

Table 1. Demographic information of each group (mean ± standard deviation).

Group

Nocturnal Sleep

Sleep Deprivation

Number

20

20

Gender

Male

Male

Age (y)

23.55±2.33

23.11±2.40

PSQI

3.54±0.93

3.66±0.50

R1-2.Was testing for anxiety carried out before inclusion in the observation group? Because anxiety also affects sleep.

Response:

Thank you very much for your suggestion. We did not use a separate anxiety questionnaire to administer to participants. However, we screened participants using the Pittsburgh Sleep Quality Index, which is widely used to test an individual's sleep quality over the last month. As you mentioned, sleep can be affected by many factors, and anxiety status is one of them. If an individual has associated anxiety symptoms, their sleep quality must be affected as well, so even if a sleep quality questionnaire is used, it is able to exclude participants with anxiety problems. Therefore if the Pittsburgh score was less than 5, we assumed that the participant did not have a sleep quality problem. Thanks again for your comments.

R1-3. How was sleep deprivation achieved? Did the patients drink caffeinated drinks?

Response:

Thank you very much for your suggestion. As mentioned in "2.3", the sleep deprivation process was conducted under the supervision of two experimenters, and caffeine and tea were not allowed to be consumed during the entire sleep deprivation period.

“During the TSD period, participants were only allowed to engage in non-strenuous ac-tivities, such as talking, reading, playing games, and using computers. Participants could move freely and perform light physical activities, but they could not leave the laboratory. Smoking and drinking stimulant drinks (e.g., coffee, tea, cola) were forbid-den during the study. The experiment was performed on two participants at a time. The researchers accompanied and observed the participants throughout the entire du-ration of the experiment to ensure that they stayed awake for the whole TSD period.”

R1-4. A large number of abbreviations are used in the figures and diagrams; for clarity, it is necessary to provide explanations for the abbreviations directly below the figures.

Response:

Thank you very much for your suggestion. We have provided additional explanations of the abbreviations directly below the figures.

Reviewer 2 Report

This study has an experimental design, where the authors have tried to see the effect of 8 hours of restorative sleep on working memory and event-related potentials. It is a nicely done study, even though the authors did not find any significant differences. I have the following concerns about the study:

1. A sleep study was done; hence, authors should provide sleep data. (N1, N2, N3 and REM sleep)

2. The task chosen to assess working memory seems sensitive to the effect of a working memory deficit after sleep deprivation in young adults. Can authors shed light on this during discussion?

3. Can authors provide statistical analysis showing differences in sleep architecture during normal and RS nights? and correlate it to working memory scores.

Thank You!

This will be good I authors can make language more fluent and structured. 

Author Response

Thank you for your letter and for the reviewers’ comments concerning our manuscript entitled “Decrease in the P2 Amplitude of Object Working Memory after 8h-Recovery Sleep following 36h-Total Sleep Deprivation: An ERP Study”. All of the comments were valuable and helpful for revising and improving our paper. We have studied comments carefully and have made revisions that we hope will meet with approval. The revised portions are marked in red in the paper. The main corrections in the paper and the responses to the reviewers’ comments are as follows:

Review#2

This study has an experimental design, where the authors have tried to see the effect of 8 hours of restorative sleep on working memory and event-related potentials. It is a nicely done study, even though the authors did not find any significant differences. I have the following concerns about the study:

R2-1.A sleep study was done; hence, authors should provide sleep data. (N1, N2, N3 and REM sleep)

Response:

Thank you very much for your suggestion. As mentioned in “2.3”: “The SD group slept all night in the laboratory and underwent sleep monitoring (ensuring a sleep time of at least 8h).” We only monitored the sleep of the SD group in the laboratory the night before SD. In fact we monitored their sleep duration using sleep bracelets and did not have data on sleep staging. We have modified the original article to avoid ambiguity in understanding. I hope the revised formulation will address your concerns.

“The SD group slept all night in the laboratory and underwent sleep monitoring by sleep bracelets (ensuring a sleep time of at least 8h).”

R2-2.The task chosen to assess working memory seems sensitive to the effect of a working memory deficit after sleep deprivation in young adults. Can authors shed light on this during discussion?

Response:

Thank you very much for your suggestion. We added clarification in the discussion, page 9:

“Studies have shown that the improvement effect of RS can also be reflected in the changes in ERPs late components [34]. Our study provides evidence for this type of conclusion from the perspective of the early components of EEG. The working memory task selected in this study is sensitive to sleep deprivation, so we can see the differ-ences in related EEG components under different states. Working memory is the abil-ity to temporarily store and manipulate information during cognitive tasks. To this day, working memory is considered the core foundation of human advanced cognitive ac-tivities [35]. The early components selected in this study reflect the early stages of at-tention and perception processing in working memory processing [36].RS first im-proves the initial stage of attention regulation reflected by N1 and P2, which is further manifested in the shortening of the P2 latency, reflecting the recovery of the individu-al’s attention needed to process stimuli in the visual cortex [37]. Combined with the results of our research, it can be seen from the changes in the P2 components before and after RS that the improvement primarily occurs in the process of attention and sensory input. With the recovery of the sensory process, the increase in the potential amplitude of the P3 component after the P2 component may reflect the recovery of the participants' ability to discern the target stimulus since P3 latency reflects the time window for stimulus classification and evaluation [38,39,40].”

R2-3.Can authors provide statistical analysis showing differences in sleep architecture during normal and RS nights? and correlate it to working memory scores.

Response:

Thank you very much for your suggestion. We did not perform sleep monitoring for normal and RS nights, so we may not have a way to provide results from this section. Your suggestion makes a lot of sense, and in fact we did consider including this section in the design of the experiment, but in the end, due to limitations of the equipment and experimental conditions, we did not collect data that would have been valuable in this section. Therefore, we also added this point into the limitation and hope that future research can solve this problem. Although this part of sleep architecture data is meaningful, it does not affect the main research focus of this study. The main intention of this study was to explore the changes in event-related potential components related to working memory before and after sleep deprivation as well as after restorative sleep deprivation using ERP techniques. And it was found that the restorative effects were mainly reflected in the enhancement of the early sensory processing stage, as well as the enhancement of perceptual matching and attentional regulation functions. Thanks again for your comments.
